METHODS AND RESOURCES

# ClipKIT: A multiple sequence alignment trimming software for accurate phylogenomic inference

**Jacob L. Steenwyk**[1]*, **Thomas J. Buida, III**[2], **Yuanning Li**[1], **Xing-Xing Shen**[3],
**Antonis Rokas**[1]*

**1** Vanderbilt University, Department of Biological Sciences, Nashville, Tennessee, United States of America,
**2** Nashville, Tennessee, United States of America, **3** Ministry of Agriculture Key Lab of Molecular Biology of
Crop Pathogens and Insects, Institute of Insect Sciences, Zhejiang University, Hangzhou, China

* jacob.steenwyk@vanderbilt.edu (JLS); antonis.rokas@vanderbilt.edu (AR)

## Abstract

Highly divergent sites in multiple sequence alignments (MSAs), which can stem from erroneous inference of homology and saturation of substitutions, are thought to negatively impact phylogenetic inference. Thus, several different trimming strategies have been developed for identifying and removing these sites prior to phylogenetic inference. However, a recent study reported that doing so can worsen inference, underscoring the need for alternative alignment trimming strategies. Here, we introduce ClipKIT, an alignment trimming software that, rather than identifying and removing putatively phylogenetically uninformative sites, instead aims to identify and retain parsimony-informative sites, which are known to be phylogenetically informative. To test the efficacy of ClipKIT, we examined the accuracy and support of phylogenies inferred from 14 different alignment trimming strategies, including those implemented in ClipKIT, across nearly 140,000 alignments from a broad sampling of evolutionary histories. Phylogenies inferred from ClipKIT-trimmed alignments are accurate, robust, and time saving. Furthermore, ClipKIT consistently outperformed other trimming methods across diverse datasets, suggesting that strategies based on identifying and retaining parsimony-informative sites provide a robust framework for alignment trimming.

## Introduction

Multiple sequence alignment (MSA) of a set of homologous sequences is an essential step of molecular phylogenetics, the science of inferring evolutionary relationships from molecular sequence data. Errors in phylogenetic analysis can be caused by erroneously inferring site homology or saturation of multiple substitutions [1], which often present as highly divergent sites in MSAs. To remove errors and phylogenetically uninformative sites, several methods "trim" or filter highly divergent sites using calculations of site/region dissimilarity from MSAs [1–4]. A beneficial by-product of MSA trimming, especially for studies that analyze hundreds of MSAs from thousands of taxa [5], is that trimming MSAs reduces the computational time

**Data Availability Statement:** All alignments and phylogenies inferred in this study will be available from figshare (doi: 10.6084/m9.figshare.12401618) upon publication. The following link is

provided for review purposes only https://figshare.com/s/bd07b70b510bca3155b9.

**Funding:** J.L.S. and A.R. were supported by the Howard Hughes Medical Institute through the James H. Gilliam Fellowships for Advanced Study program. A.R. was supported by the National Science Foundation (DEB-1442113), the Guggenheim Foundation, the Burroughs Wellcome Fund, and the National Institutes of Health / National Institute of Allergy and Infectious Diseases (1R56AI146096-01A1). X.X.S. was supported by the start-up grant from the "Hundred Talents Program" at Zhejiang University and the Fundamental Research Funds for the Central Universities. The funders had no role in study design, data collection and analysis, decision to publish, or preparation of the manuscript.

**Competing interests:** The authors have declared that no competing interests exist.

**Abbreviations:** ABS, average bipartition support; BMGE, Block Mapping and Gathering with Entropy; CI, continuous integration; GTR, general time reversible; MSA, multiple sequence alignment; nRF, normalized Robinson–Foulds; WAG, Whelan and Goldman.

and memory required for phylogenomic inference. Nowadays, MSA trimming is a routine part of molecular phylogenetic inference [6].

Despite the overwhelming popularity of MSA trimming strategies, a recent study revealed that trimming often decreases, rather than increases, the accuracy of phylogenetic inference [7]. This decrease suggests that current strategies may remove phylogenetically informative sites (e.g., parsimony-informative and variable sites) that have previously been shown to contribute to phylogenetic accuracy [8]. Furthermore, it was shown that phylogenetic inaccuracy is positively associated with the number of removed sites [7], revealing a speed–accuracy trade-off wherein trimmed MSAs decrease the computation time of phylogenetic inference but at the cost of reduced accuracy. More broadly, these findings highlight the need for alternative MSA trimming strategies.

To address this need, we developed ClipKIT, an MSA trimming algorithm based on a conceptually novel framework. Rather than aiming to identify and remove putatively phylogenetically uninformative sites in MSAs, ClipKIT instead focuses on identifying and retaining parsimony-informative sites, which (alongside other types of sites and features of MSAs, such as variable sites and alignment length) have previously been shown to be phylogenetically informative [8]. ClipKIT implements a total of 5 different trimming strategies. Certain ClipKIT trimming strategies allow users to also retain constant sites, which inform base frequencies in substitution models [9], and/or trim alignments based on the fraction of taxa represented by gaps per site (or site gappyness). We tested the accuracy and support of phylogenetic inferences using ClipKIT and other alignment trimming software using nearly 140,000 alignments from empirical datasets of mammalian and budding yeast sequences [8] and simulated datasets of metazoans, plants, filamentous fungi, and a larger sampling of budding yeasts sequences [10–13]. We found that ClipKIT-trimmed alignments led to accurate and well-supported phylogenetic inferences that consistently outperformed other alignment trimming software. Additionally, we note that ClipKIT-trimmed alignments can save computation time during phylogenetic inference. Taken together, our results demonstrate that alignment trimming based on identifying and retaining parsimony-informative sites is a robust alignment trimming strategy.

## Results

To test the efficacy of ClipKIT, we examined the accuracy and support of single-gene and species-level phylogenetic trees inferred from untrimmed MSAs and MSAs trimmed using 14 different strategies (Table 1) across 4 empirical genome-scale datasets and 4 simulated datasets. The 4 empirical datasets correspond to the untrimmed amino acid and nucleotide MSAs from 24 mammals ($N_{alignments} = 4,004$) and 12 budding yeasts ($N_{alignments} = 5,664$) [8]. The 4 simulated datasets ($N_{alignments} = 50$ alignments per dataset or 200 total) stem from simulated nucleotide sequence evolution along the species phylogeny of 93 filamentous fungi [10] and from simulated amino acid sequence evolution along the species phylogenies of 70 metazoans [11], 46 flowering plants [12], and 96 budding yeasts [13]. MSAs were trimmed using popular alignment trimming software (Table 1), generating a total of 138,152 MSAs [(4,004 mammalian + 5,664 yeast + 200 simulated MSAs) * (14 trimming strategies, including a "no trimming" strategy) = 138,152 MSAs]. However, Gblocks and Block Mapping and Gathering with Entropy (BMGE) with an entropy threshold of 0.3 were not used for performance assessment of simulated datasets because they frequently removed entire MSAs.

We found that the 14 strategies examined occupied distinct regions of feature space suggestive of substantial differences between MSAs (Fig 1). Variation in feature space was largely driven by normalized Robinson–Foulds (nRF) and average bipartition support (ABS)

**Table 1. The 14 different MSA trimming strategies tested in this study.**

| Software | MSA trimming strategies | Approach | Parameter(s) | Reference |
|---|---|---|---|---|
| ClipKIT | ClipKIT: k | Keep parsimony-informative sites | kpi mode | This study |
| | ClipKIT: kg | Keep parsimony-informative sites and remove highly gappy sites | kpi-gappy mode; remove sites with 90% gaps | |
| | ClipKIT: kc | Keep parsimony-informative and constant sites | kpic mode | |
| | ClipKIT: kcg | Keep parsimony-informative and constant sites and remove highly gappy sites | kpic-gappy mode; remove sites with 90% gaps | |
| | ClipKIT: g | Remove highly gappy sites | gappy mode; remove sites with 90% gaps | |
| BMGE | BMGE 0.3 | Remove sites with high entropy | Entropy threshold of 0.3 | [3] |
| | BMGE | | Default entropy threshold of 0.5 | |
| | BMGE 0.7 | | Entropy threshold of 0.7 | |
| Gblocks | Gblocks | Remove sites that are gap rich and highly variable | default | [1] |
| Noisy | Noisy | Predicts homoplastic sites and remove them | default | [15] |
| trimAl | trimAl: s | Remove highly gappy and variable sites | strict mode | [2] |
| | trimAl: sp | Remove highly gappy and variable sites | strictplus mode | |
| | trimAl: go | Remove highly gappy sites | gappyout mode | |
| No trimming | No trim | N/A | N/A | N/A |

Each MSA trimming strategy tested by our study, the software used, a general description of its trimming approach, its parameters, and a citation for the software used are described here.

BMGE, Block Mapping and Gathering with Entropy; MSA, multiple sequence alignment; N/A, not applicable.

measures along the first dimension and alignment length along the second dimension for both empirical and simulated datasets (S1 Fig). In empirical datasets, we found that some ClipKIT strategies removed few sites, while others removed many and, at times, the most sites (S2 Fig). Among simulated datasets, ClipKIT trimmed substantial portions of MSAs, but variation was observed across MSAs and datasets (S3 Fig). Examination of nRF and ABS values revealed that ClipKIT performed well, and at times the best, among the MSA trimming strategies tested, suggesting that phylogenetic inferences made with ClipKIT-trimmed MSAs were both accurate and well supported (S4 and S5 Figs). Finally, counter to previous evidence suggestive of a trade-off between trimming and phylogenetic accuracy [7], we found that that ClipKIT aggressively trimmed MSAs in the empirical datasets without compromising phylogenetic tree accuracy and support (S2 and S4 Figs).

To obtain a summary of overall performance, we ranked the 14 strategies' performance for each dataset using objective desirability–based integration of nRF and ABS values [14] (Fig 2). We found that the 5 ClipKIT strategies outperformed all others for amino acid sequences in the empirical mammalian dataset (Fig 2A) as well as in the simulated metazoan and flowering plant datasets (Fig 2E and 2F). Other strategies that performed well included trimAl with the "gappyout" parameter for empirical datasets and Noisy for simulated datasets [2,15]. To evaluate MSA trimming strategy performance for empirical and simulated datasets, we examined average ranks across each set of 4 datasets and found that ClipKIT strategies were among the best performing (Fig 2I and 2J). In empirical datasets, ClipKIT's gappy strategy outperformed all others followed by no trimming, trimAl with the "gappyout" parameter, and then 4 other ClipKIT strategies (Fig 2I). In simulated datasets, all strategies generally performed better than in empirical datasets; the no trimming strategy ranked best followed by all 5 ClipKIT strategies (Fig 2J). These results suggest that ClipKIT, which focuses on retaining parsimony-informative

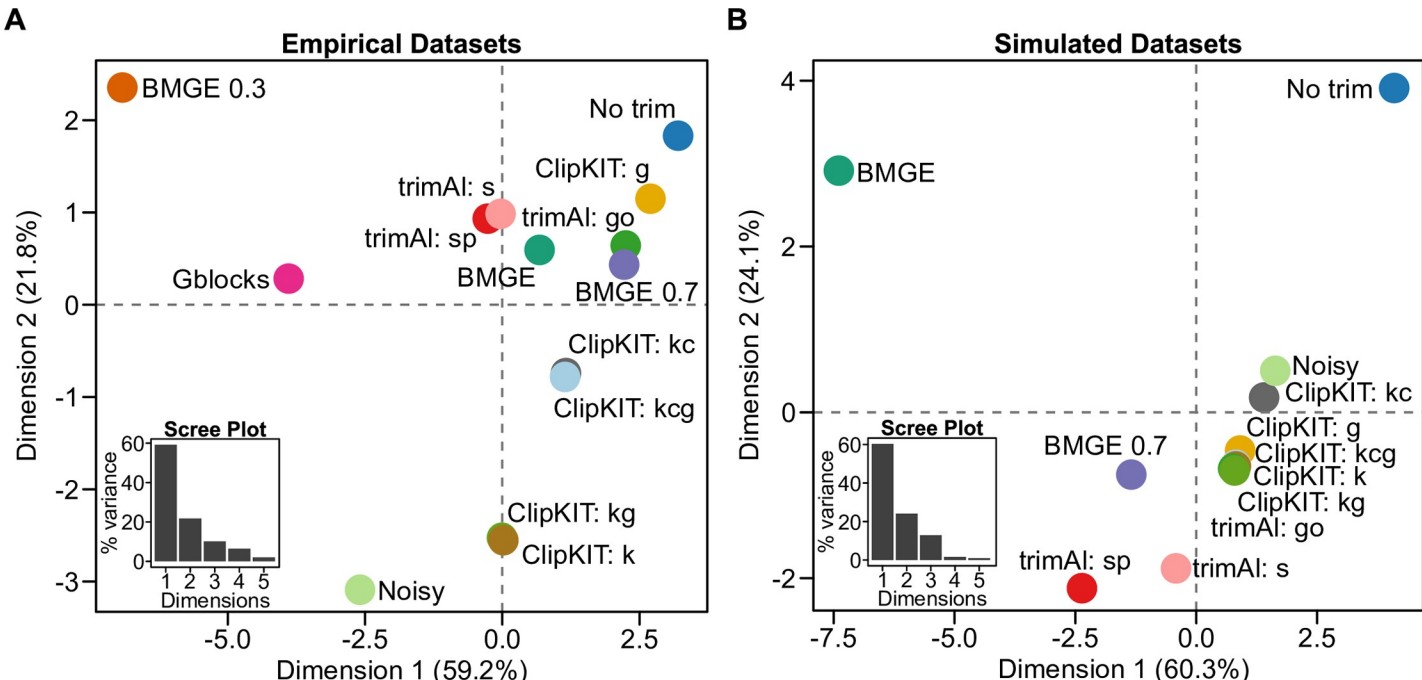

**Fig 1. The 14 alignment trimming strategies tested differ in resulting MSAs and metrics of phylogenetic tree accuracy and support.** Principal component analysis of alignment length, nRF, and ABS values across the 14 MSA trimming strategies for 4 empirical datasets (A) and 4 simulated datasets (B). Insets of scree plots depict the percentage of variation explained (y-axis) for the first 5 dimensions (x-axis). Data were scaled prior to conducting principal component analysis. Note that the BMGE 0.3 and Gblocks strategies are not represented in Fig 1B because they frequently removed entire alignments and were therefore removed from the analysis of simulated sequenced. Data used to generate this figure can be found on figshare (doi: 10.6084/m9.figshare.12401618). ABS, average bipartition support; BMGE, Block Mapping and Gathering with Entropy; MSA, multiple sequence alignment; nRF, normalized Robinson–Foulds.

sites, was on par with no trimming and frequently outperformed strategies that focus on removing highly divergent sites.

To examine the accuracy of branch lengths across MSA trimming strategies, we conducted a correlation analysis between individual branch lengths in gene trees inferred from trimmed alignments (treatment) and those inferred from untrimmed alignments (control). Because this analysis requires that untrimmed alignments are highly accurate, we conducted it only for individual simulated gene alignments. Notably, in our experimental set up, branch length estimates using trimmed alignments cannot be "more" accurate than untrimmed alignments. Thus, an alignment trimming algorithm that does not negatively influence branch length estimates will have a Spearman rank correlation coefficient of 1.0. Examination of Spearman rank correlation coefficients revealed that branch lengths of trimmed alignments were typically very highly correlated with the branch lengths of untrimmed alignments (S6–S9 Figs); ClipKIT strategies had correlation coefficients of 1.0, suggesting that branch lengths inferred using ClipKIT-trimmed alignments are accurate.

To evaluate the performance of the 14 strategies for species-level phylogenetic inference, we conducted concatenation- and quartet-based phylogenetic inference using IQ-TREE and ASTRAL v. 5.7.3 [16], respectively. We found that all strategies resulted in nearly identical and well-supported phylogenies (S10–S12 Figs). We also calculated tree certainty, an information theory-based measure of tree incongruence, which was used to summarize the degree of agreement with the reference topology across gene trees. The output from this analysis is a single value ranging from 0 to 1, where low values reflect high levels of incongruence among gene trees in the reference topology, and high values reflect low levels of incongruence with the

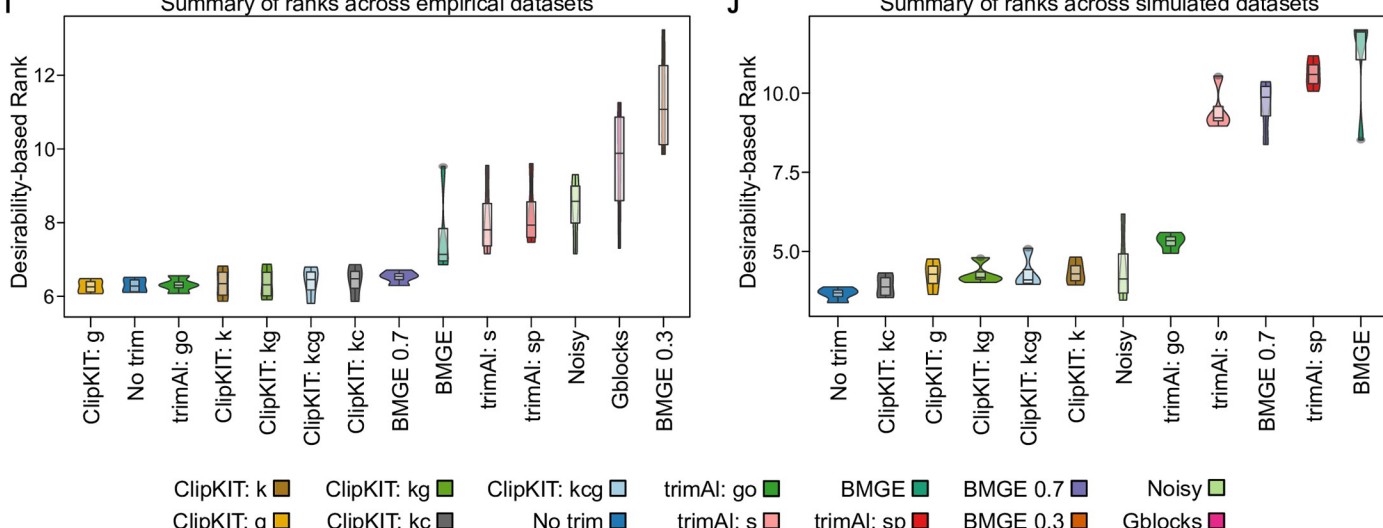

**Fig 2. ClipKIT is a top-performing software for trimming MSAs.** Desirability-based integration of accuracy and support metrics per MSA facilitated the comparison of relative performance of the 14 different MSA trimming strategies for empirical (A–D) and simulated (E–H) datasets. Examination of performance for individual datasets and average performance across empirical (I) and simulated (J) datasets revealed that ClipKIT is a top-performing software. MSA trimming strategies are ordered along the x-axis from the highest-performing strategy to the lowest-performing one according to average desirability–based rank. Boxplots embedded in violin plots have upper, middle, and lower hinges that represent the first, second, and third quartiles. Whiskers extend to 1.5 times the interquartile range. Data used to generate this figure can be found on figshare (doi: 10.6084/m9.figshare.12401618). AA, amino acid; BMGE, Block Mapping and Gathering with Entropy; MSA, multiple sequence alignment; NT, nucleotide.

reference topology among gene trees [17]. Tree certainty values were typically high and similar across all trimming strategies except for a few instances where certain strategies, which do not include ClipKIT strategies, significantly underperformed compared to all the others (S13 Fig). Among simulated datasets, we found that ClipKIT strategies reduced computation time by an average of approximately 20% compared to no trimming.

## Discussion

Current state-of-the-art MSA trimming strategies focus on the removal of highly divergent sites. Highly divergent sites are thought to lack phylogenetic signal either because they represent sites that have become mutationally saturated due to the occurrence of multiple substitutions or because they are the result of inaccurate inference of homology [18]. A previous analysis suggested that MSA trimming strategies often decreased the accuracy of phylogenetic inference [7], highlighting the need for new strategies.

To address this need, we developed ClipKIT, an alignment trimming software that focuses on identifying and retaining parsimony-informative sites. Examination of the accuracy and support of phylogenetic inferences revealed that ClipKIT strategies consistently and frequently outperformed other MSA trimming strategies and were on par with no trimming. These results suggest that MSA trimming strategies focused on retaining phylogenetically informative sites, such as parsimony-informative sites, hold promise for developing more accurate MSA trimming strategies. We anticipate that ClipKIT will be useful for phylogenomic inference and the quest to build the tree of life.

## Methods

### ClipKIT availability and usage

ClipKIT is a stand-alone software written in the Python programming language (https://www.python.org/) and is available from GitHub, (https://github.com/JLSteenwyk/ClipKIT) and PyPi (https://pypi.org/). Complete documentation is available online (https://jlsteenwyk.com/ClipKIT/). ClipKIT differs from most MSA trimming software in that it focuses on identifying and retaining parsimony-informative sites from MSAs rather than on removing highly divergent ones. To do so, ClipKIT conducts site-by-site examination of MSAs and determines whether they should be retained or trimmed based on the strategy of ClipKIT being used and how the site has been classified. During site-by-site examination of MSAs, sites are either classified as parsimony-informative, as constant sites, or neither. Note that other types of sites and features of MSAs have previously been shown to be phylogenetically informative (e.g., variable sites and MSA length); however, ClipKIT focuses on parsimony-informative sites. Parsimony-informative sites are defined as sites that contain at least 2 character states that occur in at least 2 taxa. Constant sites are defined as sites with only 1 character state that appears in at least 2 taxa [19]. Across the various ClipKIT strategies, parsimony-informative sites are always retained, constant sites are either retained or removed, and sites that are neither parsimony-informative nor constant are always removed.

Previous work [4] identified 2 types of "aberrant sites": (1) sites where only 1 sequence is represented in the alignment; and (2) sites where only 2 taxa are represented and lack homology (defined by a null model of genome-wide sequence similarity based on species-level divergences) to any other taxa in the alignment. For the first strategy, sites with these features in MSAs may stem from a genuine insertion event in 1 taxon or from assembly, annotation, and/or alignment errors; for the second strategy, homology is defined according to a null model of expected homology based on species-level sequence divergence. ClipKIT removes any sites

that are not parsimony-informative or constant, and it also removes sites that contain high percentages of gaps. Thus, such "aberrant sites" are typically removed by ClipKIT.

Lastly, ClipKIT can also perform alignment trimming based on site gappyness, which is defined as the percentage of taxa that contain a gap character state (as opposed to a nucleotide or amino acid character state) at a given site. The 5 ClipKIT trimming strategies are summarized as follows:

(1) kpi: a strategy that retains sites that are parsimony-informative, which is specified with the following command:

```
clipkit <MSA> -m kpi;
```

This strategy executes the following pseudocode:

FOR site in alignment:

>IF site is parsimony-informative

>>keep the site

>ELSE

>>remove the site

ENDFOR

(2) kpic: a strategy that retains sites that are either parsimony-informative or constant, which is specified with the following command:

```
clipkit <MSA> -m kpic;
```

This strategy executes the following pseudocode:

FOR site in alignment:

>IF site is parsimony-informative or constant

>>keep the site

>ELSE

>>remove the site

ENDFOR

(3) gappy: a strategy that removes sites that are gappy-rich (defined as sites with ≥90% gaps), which is specified with the following command:

```
clipkit <MSA> -m gappy,
```

Because gappy-based trimming is the default strategy, it can also be executed with the following command:

```
clipkit <MSA>;
```

This strategy executes the following pseudocode:

FOR site in alignment:

>IF site has <90% gaps

>>keep the site

>ELSE

>>remove the site

ENDFOR

(4) kpi-gappy: a combination of strategies 1 and 3, which is specified with the following command:

```
clipkit <MSA> -m kpi-gappy;
```

This strategy executes the following pseudocode:

FOR site in alignment:

>IF site is parsimony-informative AND has <90% gaps

>>keep the site

>ELSE

>>remove the site

ENDFOR

                                      

(5) kpic-gappy: a combination of strategies 2 and 3, which is specified with the following command:

```
clipkit <MSA> -m kpic-gappy.
```

This strategy executes the following pseudocode:

FOR site in alignment:

>IF site is (parsimony-informative OR constant) AND has <90% gaps

>>keep the site

>ELSE

>>remove the site

ENDFOR

All output files have the same name as the input files with the addition of the suffix ".clip-kit." Users can specify output files names with the -o/—output option. For example, an alignment may have the output name "ClipKIT_trimmed_aln.fa" with the following command:

```
clipkit <MSA> -o ClipKIT_trimmed_aln.fa.
```

To enable users to fine-tune alignment trimming parameters, we provide an additional option for users to specify their own gappyness threshold, which can range between 0 and 1. For example, to retain sites with <95% gaps, the following command would be used:

```
clipkit <MSA> -g 0.95
```

This gappyness threshold would execute the following pseudocode:

FOR site in alignment:

>IF site has <95% gaps

>>keep the site

>ELSE

>>remove the site

ENDFOR

In practice, we recommend the use of very high gappyness thresholds; the use of lower thresholds may remove too many sites, which may worsen phylogenetic inferences [8].

To enable users to examine the trimmed sites/regions from MSAs, we have also implemented a logging option in ClipKIT. When used, the logging option outputs an additional 4-column file with the following information: column 1, position in the alignment (starting at 1); column 2, whether or not the site was trimmed or kept; column 3, reports if the site was parsimony-informative, constant, or neither; and column 4, reports the gappyness of the site. Log files are generated using the -l/—log option:

```
clipkit <MSA> -l
```

We anticipate that this information will be helpful for alignment diagnostics, fine-tuning of trimming parameters, and other reasons.

To enable seamless integration of ClipKIT into preexisting pipelines, 8 file types can be used as input. More specifically, ClipKIT can input and output *fasta*, *clustal*, *maf*, *mauve*, *phylip*, *phylip-sequential*, *phylip-relaxed*, and *stockholm* formatted MSAs. By default, ClipKIT automatically determines the input file format and creates an output file of the same format; however, users can specify either with the -if/—input_file_format and -of/—output_file_format options. For example, an input file of *fasta* format and a desired output file of *clustal* format can be specified using the following command:

```
clipkit <MSA> -if fasta -of clustal
```

Recent analyses indicate that approximately 28% of available computational tools fail to install due to implementation errors [20]. To overcome this hurdle and ensure archival stability of ClipKIT, we implemented state-of-the-art software development practices and design principles. More specifically, ClipKIT is composed of highly modular, extensible, and reusable code, which allows for easy debugging and seamless integration of new functions and features.

We wrote a total of 118 unit and integration tests resulting in 97% code coverage. We also implemented a robust continuous integration (CI) pipeline to automatically build, package, and test ClipKIT whenever code is modified. This CI pipeline runs a testing matrix for Python versions 3.6, 3.7, and 3.8. Given the current configuration, building and testing ClipKIT for future versions of Python will be straightforward. Lastly, central ClipKIT functions rely on few dependencies (i.e., BioPython [21] and NumPy [22]). In summary, we have taken several measures to ensure ClipKIT implements MSA trimming strategies that do not sacrifice the accuracy of phylogenetic inference but also safeguard that ClipKIT will be a long-lasting computational tool for the field of molecular phylogenetics.

## Practical considerations when using ClipKIT

Although ClipKIT strategies performed well across empirical genome-scale and simulated datasets, we acknowledge that testing every possible evolutionary scenario is impossible. This is further complicated by the lack of large-scale phylogenomic data matrices in which the true evolutionary relationships among organisms are known. Therefore, we recommend using multiple trimming strategies available in ClipKIT and examining the resulting ABS values for trees. Considering high ABS values often corresponded to lower nRF values (S4 and S5 Figs), using the resulting phylogeny with the highest ABS value may be a representative of the phylogeny that most closely approximates the true evolutionary history. This may require substantially greater computation time. To potentially ameliorate the computation time issue that may arise, we recommend creating subsets of larger datasets that span alignments of various lengths and testing multiple trimming strategies on the reduced dataset.

Although constant sites are thought to be important for informing parameters of substitution models [9], we observed variation in the performance of ClipKIT strategies that retain only parsimony-informative sites (kpi and kpi-gappy) and the performance strategies that retain parsimony-informative and constant sites (kpic and kpic-gappy). More specifically, at times, strategies kpi and kpi-gappy outperformed kpic and kpic-gappy, suggesting that constant sites may not always be informative to substitution models. However, we note that trimming nucleotide sequences with strategies kpi and kpi-gappy may warrant ascertainment bias correction for nucleotide sequences because constant sites are absent from the trimmed alignments.

## Dataset acquisition and generation

To test the efficacy of strategies from ClipKIT and other alignment trimming software (Table 1), we used a total of 8 empirical and simulated datasets. For empirical datasets, we obtained publicly available untrimmed amino acid and nucleotide MSAs from 24 mammals ($N_{alignments}$ = 4,004) and 12 budding yeasts ($N_{alignments}$ = 5,664) totaling 4 datasets [8]. Publicly available amino acid alignments were generated with MAFFT, v. 7.164, using the G-INS-I strategy with a gap penalty of 1.53 [23]. Publicly available nucleotide alignments were generated by mapping nucleotide sequences onto the amino acid alignments. For simulated datasets, we simulated sequence evolution along proposed species phylogenies of 93 filamentous fungi [10] and from simulated amino acid sequence evolution along the species phylogenies of 70 metazoans [11], 46 flowering plants [12], and 96 budding yeasts [13] ($N_{alignments}$ = 50 alignments per dataset or 200 total).

Simulated sequences were generated with INDELible v. 1.03 [24] using parameters suggested by the software developers. INDELible was chosen to generate simulated sequences because of its ability to also simulate insertion and deletion events, which are represented by gaps, a common feature in MSAs. Nucleotide alignments for filamentous fungi were generated

using the general time reversible (GTR) substitution model [25]. Additional parameters specified were state frequencies values of 0.1, 0.2, 0.3, and 0.4 for T, C, A, and G nucleotides, respectively. We specified the substitution rate matrix using the scheme outlined in S1 Table. Insertion and deletion rates were set to be 5% as frequent as single substitutions. Insertion and deletions occurred according to the power law distribution (a = 1.7, M = 500). The tree's root length was set to 1,000. For amino acids, all parameters were the same except the insertion, and deletion rates were set to be 1% as frequent as single substitutions using the Whelan and Goldman (or WAG) model of substitutions, which was also used to specify state frequencies [26].

The resulting empirical and simulated MSAs were trimmed using 14 popular alignment trimming strategies (Table 1). Altogether, we generated a total of 138,152 MSAs [(4,004 mammalian + 5,664 yeast + 200 simulated MSAs) * (14 trimming strategies, including a "no trimming" strategy) = 138,152 MSAs], which were used to evaluate the performance of ClipKIT and other alignment trimming strategies.

## Measuring accuracy and support of phylogenetic inferences

Phylogenetic inferences from MSAs were made using IQ-TREE v. 1.6.11 [9]. For nucleotide sequences, we used a GTR substitution model [27] with empirical base frequencies and a discrete Gamma model with 4 rate categories [28] or "GTR+F+G"; for amino acid sequences, we used the general WAG model of substitutions [26] with empirical base frequencies and a discrete Gamma model with 4 rate categories [28] or "WAG+F+G."

Tree accuracy was measured using nRF distances as calculated by ape v. 5.3 [29] in R package (https://cran.r-project.org/), by comparing the inferred gene phylogenies to their species phylogenies. Tree support was measured using ABS from 5,000 ultrafast bootstrap approximations in IQ-TREE [30]. To determine if alignment trimming strategies resulted in substantially different alignment lengths, nRF values, and ABS values, we conducted principal component analysis using the R packages FactoMineR v. 2.3 [31] and factoextra v. 1.0.6 [32]. All plots were made with FactoMineR, factoextra, and ggplot2, v. 2.3.1 [33], in the R, 3.6.2 (https://cran.r-project.org/), programming environment.

To summarize nRF and ABS values into a single summary metric, we used desirability functions. Desirability functions rescale a distribution of values to be between 0 and 1 depending on whether or not low values (e.g., nRF) or high values (ABS) are best. More specifically, these transformations were conducted using the following approach:

For nRF values:

$$desirability_{low} = \begin{cases} 0 & Y > B \\ \dfrac{Y-A}{B-A} & A \leq Y \leq B \,, \\ 1 & Y < A \end{cases}$$

where $Y$ is the variable value, $A$ is the maximum nRF value, and $B$ is the minimum nRF value.

For ABS values:

$$desirability_{high} = \begin{cases} 0 & Y < A \\ \dfrac{Y-A}{B-A} & A \leq Y \leq B \,, \\ 1 & Y > B \end{cases}$$

where $Y$ is the variable value, $A$ is the minimum ABS value, and $B$ is the maximum ABS value.

These transformations were conducted for the 14 different trimming strategies on a per gene basis. The resulting values were used to rank the relative performance of the 14 trimming strategies.

To examine the accuracy of branch lengths among single-gene trees, Spearman rank correlations of branch lengths were calculated between untrimmed (control) and trimmed (treatment) simulated MSAs. To do so, the topologies of the untrimmed and trimmed phylogenies must be identical. Therefore, branch lengths were inferred along phylogenies that were constrained to match the reference tree topology using IQ-TREE [9]. This analysis was only done for simulated sequences because high confidence in alignment quality and true tree topology is required. Spearman rank correlations were conducted using the ggpubr v.0.2.5 [34] package in the R, 3.6.2 (https://cran.r-project.org/), programming environment.

For species-level phylogenetic inferences, we used concatenated alignments of trimmed MSAs as input to IQ-TREE [9]. Species-level phylogenetic inferences were also examined when using the quartet-based approach implemented in ASTRAL v. 5.7.3 [16], in which single-gene trees were used as input. Lastly, support among single-gene trees for references topologies was assessed using the information theory-based measure tree certainty [17,35,36], which is implemented in RAxML, v. 8.2.10 [37].

## Software availability

ClipKIT is available from GitHub (https://github.com/JLSteenwyk/ClipKIT) and PyPi (https://pypi.org/project/clipkit). Complete ClipKIT documentation is available online (https://jlsteenwyk.com/ClipKIT/).

## Supporting information

**S1 Fig. Data are well represented in principal component analysis.** Examination of the factors contributing to the variation of the 14 trimming strategies in feature space for empirical datasets (panels A–D) and simulated datasets (panels E–H). Examination of variable representation along the first and second dimensions of the principal component analysis (see Fig 1) revealed that alignment length, nRF, and ABS were well represented in empirical (A) and simulated (E) datasets. Variable correlation plots for empirical (B) and simulated (F) datasets show the relationship among variables. Broadly, we found that variable types (i.e., alignment length, nRF, and ABS) were correlated with one another across datasets. Examination of contribution of variables along the first and second dimensions for empirical datasets (C and D, respectively) as well as simulated datasets (G and H, respectively) revealed that ABS and nRF contributed the most along the first dimension and alignment length contributed the most along the second dimension. In these figures, the red dashed line represents the expected average contribution if all variables contributed equally. Abbreviations used in the figure are as follows: ABS, average bipartition support; Aln. len, alignment length; A.P., *Aspergillus* and Penicillium; F.P., Flowering plants; Met, Metazoans; nRF, normalized Robinson–Foulds; Sac, Saccharomycotina. Data used to generate this figure can be found on figshare (doi: 10.6084/m9.figshare.12401618).
(TIF)

**S2 Fig. Lengths of trimmed MSAs and the associated nRF and ABS values across empirical datasets.** For the mammalian AA and NT sequences (A and B) as well as the yeast AA and NT sequences (C and D), wide variation was observed in the fraction of the original alignment trimmed by different trimming strategies. MSA trimming strategies are ordered along the x-axis from the least aggressive trimmer to the most aggressive trimmer according to the average

fraction of the original alignment in that remained in the trimmed alignment (y-axis). Distributions of nRF (E–H) and ABS (I–L) values were subsequently examined. MSA trimming strategies are ordered along the x-axis from highest- to lowest-performing according to average nRF or ABS value. nRF and ABS values were Z transformed per gene prior to plotting their distribution. Boxplots embedded in violin plots have upper, middle, and lower hinges that represent the first, second, and third quartiles. Whiskers extend to 1.5 times the interquartile range. Note that the yeast dataset ($N$ = 12) is at the lower threshold of Noisy's recommended minimum number of sequences. Data used to generate this figure can be found on figshare (doi: 10.6084/m9.figshare.12401618). AA, amino acid; ABS, average bipartition support; MSA, multiple sequence alignment; nRF, normalized Robinson–Foulds; NT, nucleotide.
(TIF)

**S3 Fig. Lengths of trimmed MSAs and the associated nRF and ABS values across simulated datasets.** Extensive variation was observed in the fraction of the original alignment trimmed by the various MSA trimming strategies (A–D). MSA trimming strategies are ordered along the x-axis from the least aggressive trimmer to the most aggressive trimmer according to the average fraction of the original alignment that remained in the trimmed alignment (y-axis). Among the various datasets, we examined the distributions of nRF (E–H) and ABS (I–L) values. MSA trimming strategies are ordered along the x-axis from highest- to lowest-performing according to average nRF or ABS value. Prior to plotting, nRF and ABS values were Z transformed on a per gene basis. Boxplots embedded in violin plots have upper, middle, and lower hinges that represent the first, second, and third quartiles. Whiskers extend to 1.5 times the interquartile range. Data used to generate this figure can be found on figshare (doi: 10.6084/m9.figshare.12401618). ABS, average bipartition support; MSA, multiple sequence alignment; nRF, normalized Robinson–Foulds.
(TIF)

**S4 Fig. Pairwise examination of the relationship between alignment length, nRF, and ABS values among empirical datasets.** Broadly, we found that higher ABS values corresponded with lower nRF values across the various datasets (A–D). Additionally, lower nRF values were typically associated with shorter alignment lengths (E–H). The only MSA trimming strategies that did not follow this trend were the ClipKIT strategies kpic, kpic-gappy, kpi, and kpi-gappy, where the alignments were shorter than others but resulted in accurate phylogenetic inferences. Similarly, we found longer alignments were associated with higher ABS values (I–L). Again, we found that the only strategies that resulted in substantially shorter alignments, but which produced well-supported phylogenetic trees, were the ClipKIT strategies kpic, kpic-gappy, kpi, and kpi-gappy. This suggests that ClipKIT was able to trim substantial portions of alignments without compromising phylogenetic accuracy and support. Bidirectional error bars extend 1 standard deviation from the mean. Error bars cross at the average of the 2 variables being examined. Data used to generate this figure can be found on figshare (doi: 10.6084/m9.figshare.12401618). ABS, average bipartition support; MSA, multiple sequence alignment; nRF, normalized Robinson–Foulds.
(TIF)

**S5 Fig. Pairwise examination of the relationship between alignment length, nRF, and ABS values among simulated datasets.** Higher ABS values were often associated with lower nRF values (A–D). Lower nRF values were often associated with shorter alignment lengths (E–H). Longer alignments were often associated with higher ABS values (I–L). Bidirectional error bars extend 1 standard deviation from the mean. Error bars cross at the average of the 2 variables being examined. Data used to generate this figure can be found on figshare (doi: 10.6084/

m9.figshare.12401618). ABS, average bipartition support; nRF, normalized Robinson–Foulds.
(TIF)

**S6 Fig. ClipKIT branch lengths are accurate using simulated alignments from a metazoan phylogeny.** Spearman rank correlation coefficients reveal that ClipKIT is a top-performing software when assessing branch length accuracy. Hex plots depict the number of datasets for a given set of x- and y-coordinates. Trimmed alignment branch lengths are depicted along the x-axis. Control or untrimmed branch lengths are depicted along the y-axis. Perfect 1-to-1 correlations are represented by the red line. The line of best fitting using a linear model is represented by the blue line. All Spearman rank correlation coefficients are shown in the top right portion of each figure. All Spearman rank correlations were statistically significant ($p < 0.01$). Data used to generate this figure can be found on figshare (doi: 10.6084/m9.figshare. 12401618).
(TIF)

**S7 Fig. ClipKIT branch lengths are accurate using simulated alignments from a phylogeny of flowering plants.** Spearman rank correlation coefficients reveal that ClipKIT is a top-performing software when assessing branch length accuracy. Hex plots depict the number of datasets for a given set of x- and y-coordinates. Trimmed alignment branch lengths are depicted along the x-axis. Control or untrimmed branch lengths are depicted along the y-axis. Perfect 1-to-1 correlations are represented by the red line. The line of best fitting using a linear model is represented by the blue line. All Spearman rank correlation coefficients are shown in the top right portion of each figure. All Spearman rank correlations were statistically significant ($p < 0.01$). Data used to generate this figure can be found on figshare (doi: 10.6084/m9. figshare.12401618).
(TIF)

**S8 Fig. ClipKIT branch lengths are accurate using simulated alignments from a phylogeny of *Aspergillus* and *Penicillium* species.** Spearman rank correlation coefficients reveal that ClipKIT is a top-performing software when assessing branch length accuracy. Hex plots depict the number of datasets for a given set of x- and y-coordinates. Trimmed alignment branch lengths are depicted along the x-axis. Control or untrimmed branch lengths are depicted along the y-axis. Perfect 1-to-1 correlations are represented by the red line. The line of best fitting using a linear model is represented by the blue line. All Spearman rank correlation coefficients are shown in the top right portion of each figure. All Spearman rank correlations were statistically significant ($p < 0.01$). Data used to generate this figure can be found on figshare (doi: 10. 6084/m9.figshare.12401618).
(TIF)

**S9 Fig. ClipKIT branch lengths are accurate using simulated alignments from a phylogeny of Saccharomycotina yeast.** Spearman rank correlation coefficients reveal that ClipKIT is a top-performing software when assessing branch length accuracy. Hex plots depict the number of datasets for a given set of x- and y-coordinates. Trimmed alignment branch lengths are depicted along the x-axis. Control or untrimmed branch lengths are depicted along the y-axis. Perfect 1-to-1 correlations are represented by the red line. The line of best fitting using a linear model is represented by the blue line. All Spearman rank correlation coefficients are shown in the top right portion of each figure. All Spearman rank correlations were statistically significant ($p < 0.01$). Data used to generate this figure can be found on figshare (doi: 10.6084/m9. figshare.12401618).
(TIF)

**S10 Fig. Concatenated alignment lengths varied by dataset and alignment trimming strategy.** Alignment lengths of concatenated data matrices for mammals (A, B), yeasts (C, D), which are the empirical datasets, and metazoans (E), flowering plants (F), *Aspergillus* and *Penicillium* (G), and Saccharomycotina yeasts (H), which are simulated datasets, varied greatly. For example, ClipKIT with the gappy strategy trimmed the least among the empirical datasets, while Noisy often trimmed the least among simulated datasets. MSA trimming strategies are ordered along the x-axis from the ones that trimmed the most to those that trimmed the least. Data used to generate this figure can be found on figshare (doi: 10.6084/m9.figshare. 12401618). MSA, multiple sequence alignment.
(TIF)

**S11 Fig. Alignment trimming strategies resulted in nearly identical metrics of accuracy and support for species-level inferences using empirical datasets.** Species-level phylogenies were inferred using a concatenation and coalescence approaches. All phylogenies inferred received nearly full support using the concatenation (A–D) and coalescence approaches (E–H). Similarly, phylogenies inferred using the concatenation (I–L) and coalescence approaches (M–P) were accurate. Variation species-level inferences for these datasets has been previously reported. Thus, we are considering these phylogenies nearly identical. Distributions among concatenation-based metrics stem from 5 independent tree searches. Data used to generate this figure can be found on figshare (doi: 10.6084/m9.figshare.12401618).
(TIF)

**S12 Fig. Alignment trimming strategies resulted in nearly identical metrics of accuracy and support for species-level inferences using simulated datasets.** Species-level phylogenies were inferred using concatenation and coalescence approaches. Nearly full support was observed for phylogenies inferred using the concatenation (A–D) and coalescence approaches (E–H). Low nRF values indicate the concatenation (I–L) and coalescence approaches (M–P) inferred accurate species-level phylogenies. Concatenation-based metrics are derived from a single tree search. Data used to generate this figure can be found on figshare (doi: 10.6084/m9. figshare.12401618).
(TIF)

**S13 Fig. Alignment trimming strategies have similar tree certainty values.** Tree certainty values were calculated using all single-gene trees and the reference tree as input. (A–D) Across empirical datasets, tree certainty values were similar across alignment trimming strategies with the exception of Gblocks, Noisy, and BMGE with an entropy threshold of 0.3 that frequently had lower tree certainty values compared to other alignment trimming strategies. (E–H) Among simulated datasets, a similar pattern was observed except BMGE often had a lower tree certainty value compared to other alignment trimming strategies. Data used to generate this figure can be found on figshare (doi: 10.6084/m9.figshare.12401618). BMGE, Block Mapping and Gathering with Entropy.
(TIF)

**S1 Table. GTR substitution rate matrix.** Simulated NT alignments were generated using this substitution rate matrix. GTR, general time reversible; NT, nucleotide.
(XLSX)

## Author Contributions

**Conceptualization:** Jacob L. Steenwyk, Yuanning Li, Xing-Xing Shen, Antonis Rokas.

**Data curation:** Jacob L. Steenwyk.

**Formal analysis:** Jacob L. Steenwyk.

**Funding acquisition:** Jacob L. Steenwyk, Antonis Rokas.

**Investigation:** Jacob L. Steenwyk, Yuanning Li, Xing-Xing Shen.

**Methodology:** Jacob L. Steenwyk, Thomas J. Buida, III, Xing-Xing Shen.

**Project administration:** Jacob L. Steenwyk, Antonis Rokas.

**Resources:** Jacob L. Steenwyk, Xing-Xing Shen, Antonis Rokas.

**Software:** Jacob L. Steenwyk, Thomas J. Buida, III.

**Supervision:** Antonis Rokas.

**Validation:** Jacob L. Steenwyk, Thomas J. Buida, III.

**Writing – original draft:** Jacob L. Steenwyk, Antonis Rokas.

**Writing – review & editing:** Yuanning Li, Xing-Xing Shen, Antonis Rokas.

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
