## [Editor Report · Decision Letter 0]

1 Jul 2020

Dear Dr Rokas, 

Thank you for submitting your manuscript entitled "ClipKIT: a multiple sequence alignment-trimming algorithm for accurate phylogenomic inference" for consideration as a Methods and Resources by PLOS Biology.

Your manuscript has now been evaluated by the PLOS Biology editorial staff, as well as by an academic editor with relevant expertise, and I'm writing to let you know that we would like to send your submission out for external peer review.

Please re-submit your manuscript within two working days, i.e. by Jul 03 2020 11:59PM.

Kind regards,

Roli Roberts

Senior Editor

PLOS Biology

---

## [Decision Letter · Decision Letter 1]

21 Aug 2020

Dear Dr Rokas,

Thank you very much for submitting your manuscript "ClipKIT: a multiple sequence alignment-trimming algorithm for accurate phylogenomic inference" for consideration as a Methods and Resources at PLOS Biology. Your manuscript has been evaluated by the PLOS Biology editors, an Academic Editor with relevant expertise, and by four independent reviewers.

You’ll see that three of the reviewers (#1, #2, #4) find your method useful, but rev #1 thinks that you need to improve the description of what it actually does (in contrast to other approaches); we agree, and think that this is doubly important for our broader readership. Several of the reviewers have further technical concerns, some of which may need additional analysis.

In light of the reviews (below), we will not be able to accept the current version of the manuscript, but we would welcome re-submission of a much-revised version that takes into account the reviewers' comments. We cannot make any decision about publication until we have seen the revised manuscript and your response to the reviewers' comments. Your revised manuscript is also likely to be sent for further evaluation by the reviewers.

We expect to receive your revised manuscript within 2 months. 

**IMPORTANT - SUBMITTING YOUR REVISION**

*Re-submission Checklist*

*Published Peer Review*

*PLOS Data Policy*

*Blot and Gel Data Policy*

Sincerely,

Roli Roberts

Senior Editor,

rroberts@plos.org,

PLOS Biology

REVIEWERS' COMMENTS:

Reviewer #1:

General comments:

The study by Steenwyk addresses an important and unappreciated issue in genomics, that sequence alignments have problems, which, if not fixed, can negatively impact downstream studies, such as generating a phylogenetic tree. They came up with a solution for phylogeny that identifies and saves phylogenetically informative sites of a multiple sequence alignment, as opposed to deleting sites that are not uninformativ. They performed a thorough comparison of multiple methods, and find that their ClipKIT methods out performs the other methods. Overall, I think the study has made important advances. But, I have some concerns that I feel need to be addressed.

The authors present one view on issues with alignments, but there are really two: 1) Determining which sites are phylogenetically informative, as ClipKIT is designed to do; 2) Removing errors in the alignment, which some other tools are designed to do, but it is not clear that ClipKIT is designed to remove errors? Such errors include over-alignment (non-homologous sequences that should not be aligned but were aligned) and under-alignment (homologous sequences that should be aligned but were not aligned). These errors too affect phylogenetic inference and many other analyses. The authors do not seem to address how ClipKIT handles such errors? In a study of alignment of 48 bird genomes, Jarvis et al 2014 (Science) developed alignment filtering tools (described in the supplement) to remove such errors before they inferred a phylogeny. Does ClipKIT work like these?

A big issue is that the authors are not clear enough on what ClipKIT does. I looked in the methods and in their online website about it. Do the authors have equations that they programmed? What criterion is used to determine a site is phylogenetically informative? What does constant sites mean (identical sequence in all species)? What does ClipKIT do with the non-phylogenetically informative sites 

Also what type of alignment software was used to make the alignments? Different alignment algorithms give more or less errors in the alignment (e.g. Jarvis et al 2014). 

After the authors demonstrate that ClipKIT outperforms maintaining phylogenetic sites, they infer phylogenetic trees comparing the trimmed alignments from multiple methods, and find almost identical, well-supported phylogenies with all methods. So, then what is the need of ClipKIT? This makes me wonder, if their simulated alignments reflected real world situations?

Overall, the paper is not prepared well enough. So, it is hard to tell issues due to missing information, or a flaw in design of the analyses.

Specific comments:

Page and line numbers need to be included, otherwise writing a review is more difficult. 

If ClipKIT is suppose to identify and retain phylogenetically informative sights, the term itself implies removal of sites.

In the introduction, the authors need to mention the primary goal of many trimming/filtering tools is to remove alignment errors. They should describe what ClipKIT does with such errors.

The authors are inconsistent in stating the number of trimming approaches used; 13, 14, 6. In table 1, I count 6 trimming approaches, 12 subapproaches, plus a no trimming control. They need to be consistent. 

nRF and ABS need to be spelled out on first use in the paper, in the results section.

The authors state that Gblocks and BMGE trimming was not evaluated on simulated data sets, because they could remove entire alignments. However, values from these methods are in the figures, including in the simulated data sets of Figure 1B.

Where is the evidence for the following sentence? "Finally, counter to previous evidence suggestive of a trade-off between trimming and phylogenetic accuracy [6], we found that ClipKIT aggressively trimmed MSAs in the empirical datasets without compromising phylogenetic tree accuracy and support."

The following sentence in the discussion seems contradictory to the finding in the results that ClipKIT trimming did not increase or decrease the accuracy of the phylogeny. "A previous analysis suggested that MSA trimming methods often decreased the accuracy of phylogenetic inference [6], highlighting the need for alternative alignment trimming approaches."

Reviewer #2:

Trimming MSA in phylogenomics

# Summary

This study describes a new approach to trimming multiple sequence alignments (MSAs) for phylogenomics. While most available methods focus on identifying and trimming putatively phylogenetically-uninformative sites, the new method (ClipKIT) aims to identify and retain phylogenetically-informative sites. The authors demonstrate the accuracy of ClipKIT with multiple empirical and simulated datasets using split-based distance metrics for phylogenetic trees. Overall, the method performs extremely well (as judged by the metrics used) and sometimes is on par with no-trimming, yet it saves computation time.

# Assessment

In general, there are no major issues with this study. It is a clever idea to focus on site retention instead of removal. The rationale and implementation of the algorithm are clear and well-justified (though I wish the pseudo-code was provided in the Suppl. Info). Speed and accuracy are a major improvement over alternative approaches indicating that ClipKIT will likely be appealing to most empirical phylogenomicists. 

My biggest quarrel with the study is the use of split-based distance metrics like the RF distance to assess accuracy (and thus making the PCAs not easily interpretable). It is well known that RF is a highly biased and sometimes problematic metric, even if normalized. However, I am not sure what would be reasonable to ask the authors to do here given the large numbers of MSAs they examined. On one hand, it would be useful at least to provide raw, as opposed to normalized RF. On the other hand, it would be better to use alternative metrics of tree comparison. Not sure if the information theoretic-based metrics developed by the senior author (tree certainty (TC) and AllTC) would be a good addition to the ms. Alternatively, using information theoretic-based RF (see https://academic.oup.com/bioinformatics/article/doi/10.1093/bioinformatics/btaa614/5866976) could be more informative. I know this may be a lot of work, and hence I am not sure what would be the best way to proceed. Results may not change, but I worry about describing the accuracy of a new method using a metric that is highly biased like RF (and normalized RF). ABS is not significantly better. 

A second issue is that MSA site trimming may not only affect topology but also branch lengths, an equally important parameter in empirical phylogenomics and molecular evolution. Comparing branch lengths and defining null models can also be tricky, but I wonder if the authors have any insight here. 

Addressing or at least discussing these two issues may help improve an already excellent, well-written manuscript.

Reviewer #3:

In this work, Steenwyk and collaborators developed an alignment-trimming algorithm aiming to identify phylogenetically-informative sites.

The authors performed tested the effectiveness under different condition (simulated and not) and showed that consistently outperformed other trimming methods across diverse datasets. 

I have some significant concern about this work:

1) The software removes parsimonious uninformative sites; however, the same sites analysed under ML and Bayesian framework can be informative.

2) Removing the parsimonious uninformative sites affect the branch length and potentially the topology.

3) It is possible to remove parsimonious uninformative sites also gaps in MEGA. However, ClipKIT allows the user to perform this using a command line, but this does not justify a publication on Plos Biology.

Reviewer #4:

The submission addresses the issue of alignment trimming for phylogenetic inference. This is a routine step in phylogenetic inference, and some of the trimming software have thousands of citations. Recently, the benefit of alignment trimming has been called into question. The software introduced here, ClipKIT, purports to address this issue. 

The work is generally well done, clearly written, and well illustrated. I only have a few comments.

Major:

- The use of "Desirability-based integration of accuracy and support metrics" makes it easier to rank the methods and summarise the results, but makes it harder to interpret the differences. Please include a precise mathematical definition of the compound measure.

Minor:

- Abstract: "Phylogenies inferred from ClipKIT-trimmed alignments are accurate, robust, and time-saving". This statement is too absolute, particularly the accuracy claim (e.g. if the input alignment is fundamentally flawed, trimming alone cannot possibly turn it into something "accurate"). It could however be said that contrary to other methods, the trees don't worsen after ClipKit trimming, and the method saves time.

- Define ABS (average bootstrap support?) and nRF as normalised Robinson-Foulds.

- In Supplementary figure 4, I find the z-score transform confusing. I would be interested to see how the ABS and nRF values compare for the different methods. The z-score makes things less interpretable. For one thing, over what population were the mean and variance computed? (across all methods and datasets? across all datasets separately for each method?)

---

## [Decision Letter · Decision Letter 2]

26 Oct 2020

Dear Dr Rokas,

Thank you for submitting your revised Methods and Resources paper entitled "ClipKIT: a multiple sequence alignment-trimming software for accurate phylogenomic inference" for publication in PLOS Biology. I have now obtained advice from two of the original reviewers and have discussed their comments with the Academic Editor. 

Based on the reviews, we will probably accept this manuscript for publication, assuming that you will modify the manuscript to address the remaining points raised by the reviewers. Please also make sure to address the data and other policy-related requests noted at the end of this email.

IMPORTANT: Please attend to the following:

a) Please address the remaining concerns raised by reviewer #1.

b) Please address my Data Policy requests (see further down).

We expect to receive your revised manuscript within two weeks. Your revisions should address the specific points made by each reviewer. In addition to the remaining revisions and before we will be able to formally accept your manuscript and consider it "in press", we also need to ensure that your article conforms to our guidelines. A member of our team will be in touch shortly with a set of requests. As we can't proceed until these requirements are met, your swift response will help prevent delays to publication.

- a cover letter that should detail your responses to any editorial requests, if applicable

*Copyediting*

*Published Peer Review History*

*Early Version*

Sincerely,

Roli Roberts

Senior Editor,

rroberts@plos.org,

PLOS Biology

DATA POLICY:

Many thanks for depositing your alignments and phylogenies in Figshare and making your code available on Github. However, we also ask that all individual quantitative observations that underlie the data summarized in the figures and results of your paper be made available in one of the following forms:

Regardless of the method selected, please ensure that you provide the individual numerical values that underlie the summary data displayed in the following figure panels as they are essential for readers to assess your analysis and to reproduce it: Figs 1, 2, S1-S13. NOTE: the numerical data provided should include all replicates AND the way in which the plotted mean and errors were derived (it should not present only the mean/average values).

REVIEWERS' COMMENTS:

Reviewer #1:

[identifies himself as Erich Jarvis)

The authors were very responsive to the reviews, and this led to a significant improvement in the manuscript. I just have a few conceptual concerns in how things are presented, that can easily be fixed with changes in the text. 

The authors were responsive to my comments about the difference between ClipKIT identifying and removing natural phylogeneticaly un-informative sites versus removing sites due to alignment errors. They however did not clearly state this in the main text, nor state the difference on phylogenetic inference. This needs to be clearly stated for the theory and the proportion of sites with suspected uninformative site and alignment errors, when possible.

The authors seem to present a contradictory message on alignment filtering strategies that are meant to improve phylogenetic inference; when some don't actually make any change at all or sometimes make the phylogeny worse. But ClipKIT trimming is suppose to make the phylogenetic inference "better" or make no change in an already accurate phylogeny. The contradictions to this view is that in response to reviewer 3, not removing the uninformative sites does not change anything; and removing the uninformative sites does not change the branch lengths. But isn't the point of ClipKIT is that by removing the uninformative sites the phylogenic inference is should improve? The branch lengths should become more accurate? Isn't removing the sequence alignment errors suppose to improve phylogenetic inference? I believe the authors show improvements, but I also think they are unnecessarily trying to play two sides of the same coin - no change or improvement in phylogeny -. The definition of no change also means no improvement. 

Lines 157-159. The reason for highly divergent sites could be more than them not being natural mutations that are not phylogenetically formative, but because some of them are due to alignment errors. Actually, I think the later reason is more likely for many highly divergent sites in the alignment. This should be mentioned.

Reviewer #2:

The authors have addresses all my concerns and further provided more nuance to the manuscript in other sections. This is an excellent contribution to the field. I personally look forward to start using ClipKIT!

---

## [Editor Report · Decision Letter 3]

10 Nov 2020

Dear Dr Rokas,

On behalf of my colleagues and the Academic Editor, Andreas Hejnol, I am pleased to inform you that we will be delighted to publish your Methods and Resources in PLOS Biology. 

PRODUCTION PROCESS

Before publication you will see the copyedited word document (within 5 business days) and a PDF proof shortly after that. The copyeditor will be in touch shortly before sending you the copyedited Word document. We will make some revisions at copyediting stage to conform to our general style, and for clarification. When you receive this version you should check and revise it very carefully, including figures, tables, references, and supporting information, because corrections at the next stage (proofs) will be strictly limited to (1) errors in author names or affiliations, (2) errors of scientific fact that would cause misunderstandings to readers, and (3) printer's (introduced) errors. Please return the copyedited file within 2 business days in order to ensure timely delivery of the PDF proof. 

If you are likely to be away when either this document or the proof is sent, please ensure we have contact information of a second person, as we will need you to respond quickly at each point. Given the disruptions resulting from the ongoing COVID-19 pandemic, there may be delays in the production process. We apologise in advance for any inconvenience caused and will do our best to minimize impact as far as possible.

EARLY VERSION

PRESS 

Kind regards,

Vita Usova

Publication Assistant, 

PLOS Biology

on behalf of

Roland Roberts,

Senior Editor

PLOS Biology